# Protective Effect of Extra Virgin Olive Oil on Cancers, Gastrointestinal Cancers, and All-Cause Mortality: A Competing Risk Analysis in a Southern Italian Cohort

**DOI:** 10.3390/cancers16213575

**Published:** 2024-10-23

**Authors:** Caterina Bonfiglio, Rosa Reddavide, Anna Maria Cisternino, Angelo Campanella, Luigi Fontana, Gianluigi Giannelli

**Affiliations:** 1Unit of Data Science, National Institute of Gastroenterology—IRCCS “Saverio de Bellis”, Castellana Grotte, 70013 Bari, Italy; angelocampanella7@gmail.com; 2Ambulatory of Clinical Nutrition, National Institute of Gastroenterology—IRCCS “Saverio de Bellis”, Castellana Grotte, 70013 Bari, Italy; rosa.reddavide@irccsdebellis.it (R.R.); annamaria.cisternino@irccsdebellis.it (A.M.C.); 3Charles Perkins Centre, Faculty of Medicine and Health, University of Sydney, Sydney, NSW 2050, Australia; luigi.fontana@sydney.edu.au; 4Department of Endocrinology, Royal Prince Alfred Hospital, Sydney, NSW 2050, Australia; 5Scientific Direction, National Institute of Gastroenterology—IRCCS “Saverio de Bellis”, Castellana Grotte, 70013 Bari, Italy; gianluigi.giannelli@irccsdebellis.it

**Keywords:** EVOO, gastrointestinal cancer, competing risk, survival

## Abstract

This study involved 2745 participants from a southern Italian cohort. The objective was to investigate the link between the consumption of extra virgin olive oil and the risk of mortality from all causes and, in particular, from gastrointestinal cancers and other cancers. The study found that individuals who consumed more than 50 g of extra virgin olive oil per day had a 60 percent lower risk of death from gastrointestinal cancers than those who consumed less than 30 g per day. Furthermore, daily consumption of 50 g of extra virgin olive oil was associated with a 50 percent lower risk of death from other cancers compared to the lowest consumption category. This study concluded that higher olive oil consumption in this cohort of Italian Mediterranean adults was linked to a reduced risk of cancer and all-cause mortality, probably due to the health benefits of the Mediterranean diet.

## 1. Introduction

The Mediterranean diet is widely recognized as a healthy dietary pattern with protective effects against several chronic diseases, including cancer [1,2,3]. This diet is characterized by a high intake of plant-based foods, fish, and the use of extra virgin olive oil (EVOO) as the primary source of fat, which is believed to contribute to its antioxidant, anti-inflammatory, and anticancer properties [1,4,5,6]. However, epidemiological evidence, particularly concerning cancer mortality, remains limited and inconsistent [7,8].

The potential anticancer effects of extra virgin olive oil (EVOO) are mediated through complex mechanisms, including epigenetic, proteomic, and metagenomic changes, with its consumption enhancing these processes via its bioactive compounds, such as monounsaturated fatty acids, antioxidant vitamins, carotenoids, terpenoids, and polyphenols (e.g., hydroxytyrosol, oleuropein, oleocanthal, pinoresinol, luteolin) [9,10]. Additionally, EVOO positively influences gut microbiota composition, promoting mucosal immune health and reducing gut permeability in both rodents and humans [11,12].

Although numerous studies have explored the link between olive oil and cancer risk, few cohort studies—primarily from Mediterranean countries—have investigated its impact on cancer mortality. A meta-analysis of these studies suggested a potential inverse relationship between olive oil consumption and cancer mortality, though the results lacked strong statistical significance [4]. Furthermore, a study from a large U.S. population, where olive oil intake is generally lower than in Mediterranean countries, also found a reduced risk of cancer death associated with higher olive oil consumption [6].

This study aims to explore the association between EVOO consumption and mortality from all causes, including gastrointestinal and other cancers, using data from a large prospective cohort in southern Italy, with follow-up from January 2005 to December 2023.

## 2. Materials and Methods

Details of the study population have been described previously [13]. Briefly, the MICOL Study [14] is a population-based prospective cohort study initiated in 1985, randomly selected from the electoral rolls of Castellana Grotte (≥30 years) and followed through three follow-ups: 1992–1993, 2005–2006, and 2017–2019. This analysis focuses on the second follow-up (2005–2006), during which a random sample of individuals aged 30–50 years (PANEL Study) was added to the original cohort to account for its aging. Out of 3614 eligible participants, 2970 (82.2% response rate) provided written informed consent to participate. We excluded 225 participants who did not complete the diet questionnaires (see Figure 1).

### 2.1. Data Collection

Study participants were interviewed to collect information on socio-demographic characteristics, health status, personal medical history, and lifestyle factors, including tobacco use (never, current) and food intake. Educational level, employment status, and marital status were noted. The data collected are shown in Table 1.

Weight was measured in underwear using a SECA^®^ electronic balance, accurate to 0.1 kg, and height was recorded with a SECA^®^ wall-mounted stadiometer, rounded to the nearest cm. Blood pressure was measured following international guidelines [15], and the average of three readings was used. Dietary intake was assessed using a validated Food Frequency Questionnaire, completed with assistance from qualified nutritionists [16]. Fasting venous blood samples were collected, with one aliquot stored at –80 °C and another used for biochemical analyses in our central laboratory.

### 2.2. Outcome Assessment

Participants were tracked until December 31, 2023. Vital status and emigrations were verified through the registry office of Castellana Grotte. Causes of death from 2006 to December 2023 were obtained from the Apulian Regional Registry using death certificates according to the WHO International Classification of Diseases (Tenth Edition). Causes of death were categorized as Gastrointestinal Cancer (GC) (codes C14-C26), Other Cancers (OCr) (ICD-10 C01-C13, C30-C97), and Deaths from Other Causes (DOC), including all other ICD-10 codes.

### 2.3. Exposure Assessment

EVOO consumption was categorized into three groups: <30 g/day, 30–50 g/day, and >50 g/day.

### 2.4. Statistical Analysis

Data are presented as mean (±SD), median (IQR) for continuous data, or frequency (%) for categorical data. Kolmogorov and Pearson’s chi-square tests were used to test differences between means and proportions. Observation time extended from enrollment until death, migration, or the end of the study (31 December 2023), whichever occurred first.

EVOO consumption was analyzed in two sets of categories: Model a: <30 g/day, 30–50 g/day, and >50 g/day, or alternatively Model b: <30 g/day, 30–40 g/day, and >40 g/day.

Since age is the most important risk factor for death, acting as a proxy for many unknown factors, we chose age at death as the time scale. We set 90 years of age as the maximum observation age to reduce comorbidity-related problems in the elderly in the more advanced age groups.

Schoenfeld residuals were used to test the proportional hazard assumption (see Appendix A). Cox proportional hazards regression, with age as the underlying time metric, were used to estimate hazard ratios (HRs) and 95% confidence intervals (CIs) for the association between EVOO consumption and all-cause mortality. The models were adjusted for sex, calorie intake excluding those from oil, Alanine Aminotransferase (ALT), waist circumference, Diastolic and Systolic Blood Pressure, relative Mediterranean Diet Score without EVOO, smoking status, and Charlson Comorbidity Index (CII). The lowest EVOO category (<30 g/day) was the reference group.

The choice of confounders was made through a careful study of already published papers on the subject. Akaike’s criterion (AIC) and Schwarz’s criterion of Bayesian information (BIC) were run to include or not include each variable in the final model [17].

Flexible parametric survival models with a competing risk approach were used to estimate Subdistribution Hazard Ratios (SHRs) for the risk of Gastrointestinal Cancer, Other Cancers, and deaths from other causes [18]. We constructed two multivariate models that included age (time scale) and EVOO consumption (Model a or alternatively Model b) adjusted for the variables mentioned above. We fitted the cause-specific cumulative incidence function (CIF) using post-estimation tools.

All statistical analyses were performed using Stata Statistical Software version 18.0 (StataCorp, College Station, TX, USA).

## 3. Results

### 3.1. Participant Characteristics

During the observation period of 16,192.905 person-years, a total of 693 participants (25.2%) died, resulting in an incidence rate of 42.61 per 1000 person-years. Among these deaths, 75 (10.8%) were due to gastrointestinal cancers: 25 from liver cancer, 25 from pancreatic cancer, and 18 from colorectal cancer. An additional 116 deaths (16.7%) were attributed to other cancers, including 31 from lung cancer, 13 from breast cancer, and 13 from prostate cancer. The remaining 502 deaths (72.4%) were due to other causes, with 13 from Alzheimer’s disease and 37 from dementia.

Table 1 presents the main characteristics of the 2745 participants, categorized by EVOO consumption.

Among them, 1277 consumed less than 30 g of EVOO per day, 914 consumed between 30 and 50 g per day, and 554 consumed more than 50 g per day. The mean daily EVOO consumption for the entire cohort was 34.39 g (±19.26 g). Participants in the highest EVOO consumption category (>50 g/day) were older (mean age: 63.37 ±13.13 years), whereas those in the lowest consumption category (<30 g/day) were younger (mean age: 49.24 ± 23.80 years). The highest EVOO consumption group had the lowest total cholesterol (TC) and LDL-cholesterol values (TC: 197.77 ± 39.08 mg/dL, LDLc: 119.86 ± 32.78 mg/dL). Among the study participants, those consuming more than 50 g of EVOO daily had lower intakes of cereals (257.9 ± 129.9 g), meat (71.6 ± 48.4 g), dairy products (661.9 ± 537.3 g), and fruit (35.8 ± 29 g), as shown in Table 2. Conversely, this group consumed more vegetables (266.2 ± 195.3 g), pulses (34.6 ± 29.4 g), wine (180.7 ± 228.9 mL), and alcohol (206.5 ± 263.2 mL). Appendix A presents the mean (SD) values for anthropometric measurements, blood pressure, and blood parameters by daily EVOO consumption category while Appendix A shows the distribution of food groups consumption by Status (Alive or Causes of Death).

### 3.2. EVOO Consumption and Cancer Mortality

As shown in Table 3, individuals consuming 30–50 g of EVOO daily had a 24% lower risk of death from all causes [HR 0.77; 95% CI (0.63; 0.93)] compared to those consuming less than 30 g daily. Those consuming more than 50 g daily had a 20% lower risk [HR 0.80; 95% CI (0.65; 0.98)], highlighting the protective effect of EVOO on overall mortality. The daily consumption of more than 50 g of EVOO for gastrointestinal cancers was linked to a 60% reduction in death risk [SHR 0.39; 95% CI (0.21; 0.73)] compared to those consuming less than 30 g daily. Similarly, those consuming more than 50 g daily had a 50% lower risk of death from other cancers [SHR 0.50; 95% CI (0.31; 0.81)]. When comparing EVOO consumption categories, those consuming 30–50 g daily had a Sub-Hazard Ratio (SHR) of 0.74 [95% CI (0.58; 0.95)] for all-cause mortality. Higher EVOO consumption, particularly above 50 g per day, was associated with reduced gastrointestinal and other cancer risks. For gastrointestinal cancers, the SHR for consumption above 50 g/day was 0.39 [95% CI (0.21; 0.73)], compared to 0.55 [95% CI (0.33; 0.93)] for those consuming above 40 g/day. For other cancers, the SHR for consumption above 50 g/day was 0.50 [95% CI (0.31; 0.81)], compared to 0.56 [95% CI (0.36; 0.85)] for those consuming above 40 g/day. No significant differences were found in other EVOO consumption categories.

No changes in risk due to gender were observed across the categories of oil consumption. The competitive risk analysis showed that adjusting for gender did not alter the magnitude of the HR in the exposure variable (see Appendix A).

The impact of EVOO consumption on gastrointestinal cancer mortality becomes evident from age 65 (Figure 2), while its effect on other cancers is noticeable from age 55 (Appendix A).

At age 80, the probability of death from gastrointestinal cancer was 1.8% for those consuming more than 50 g/day of EVOO, 3.1% for those consuming 30–50 g/day, and 3.7% for those consuming less than 30 g/day. By age 90, the probabilities increased to 3.6%, 6.1%, and 7.3%, respectively. Figure 2 indicates that individuals consuming more than 50 g/day of EVOO had a lower risk of death compared to those consuming 40–50 g/day, demonstrating a clear protective trend for higher EVOO consumption. Appendix A compares cumulative incidence functions for other cancers based on EVOO consumption categories.

## 4. Discussion

This retrospective study provides compelling evidence supporting the protective effect of extra virgin olive oil (EVOO) against cancer mortality within a Mediterranean dietary context. Our findings suggest that higher daily consumption of EVOO is associated with a significant reduction in both all-cause and cancer-specific mortality. Specifically, individuals consuming more than 50 g/day of EVOO experienced a 20% lower risk of all-cause mortality and a 60% reduction in mortality from gastrointestinal cancers compared to those with lower EVOO intake. The benefits of EVOO consumption were evident from age 65 for gastrointestinal cancers and from age 55 for other cancers.

Previous research has supported the favorable association between olive oil and reduced cancer risk, though only a few cohort studies have specifically examined its impact on cancer mortality [19,20,21]. These studies generally found a notable reduction in cancer death risk with daily olive oil consumption. Our findings align with these studies and recent meta-analyses, which reported that the highest olive oil intake was associated with a 17% lower risk of all-cause mortality [22,23]. Our data also corroborate these findings, showing a 34% reduction in all-cause mortality risk with EVOO consumption exceeding 40 g/day.

Emerging evidence suggests that regular EVOO utilization may protect against colorectal cancer (CRC) through various biological mechanisms, including microbiota modulation, inflammation reduction, and epigenetic changes. A recent study demonstrated that a high-fat diet with 60% of calories from EVOO altered the gut microbiota in mice, which was linked to CRC prevention by increasing the Firmicutes/Bacteroidetes ratio and maintaining Akkermansia populations [11]. The health benefits of EVOO are attributed to its bioactive compounds, such as hydroxytyrosol, oleuropein, oleocanthal, apigenin, and luteolin, which influence gene expression and epigenetic mechanisms [9,10]. For example, hydroxytyrosol has increased DNA methylation and decreased ENDRA expression in Caco-2 cells [24]. Recent reviews have further explored how EVOO and its phenolic compounds specifically combat CRC through various molecular pathways [10,21].

The EVOO consumed by the inhabitants of the areas where the study was conducted is produced locally, given the conspicuous presence of centuries-old olive groves in these areas. Harvesting methods are also of ancient origin and are manual, and involve the detachment of the fruit still attached to the plant, through facilitators and harvesting nets, when the drupes have reached 15% of veraison (early October) [25]. Once harvested, the olives must be transported to the mill in a short time. Delay in processing can cause oxidation and fermentation, compromising the quality of the oil. It is essential that the olives are processed within 48 h of harvesting to maintain freshness and preserve their nutritional and organoleptic properties. The best method is to transport the olives in ventilated crates. [26,27]

These procedures guarantee the production of a superior-quality oil. The olives are processed within 8 h of harvesting in the oil mill. The extraction procedure is performed strictly ‘cold’ and with mechanical systems, so as not to alter the characteristics of the product and to preserve the content of bioactive compounds. With regard to local cultivars, the area produces the Cima di Mola and Simone varieties. Recent NMR investigations have shown that the polyphenol content of the Cima di Mola cultivar is superimposable on the better-known Coratina variety, known for its particularly high content of phenolic compounds with biological activity. The profile of phenolic compounds for the Simone cultivar is similar but more variable [28,29].

Our study found that the highest EVOO consumption category was associated with older participants who adhered well to the Mediterranean diet. These individuals consumed more vegetables and legumes and less cereal, meat, dairy, fruit, and fish than other groups. Those consuming more than 50 g/day of EVOO experienced lower incidences of gastrointestinal and other cancers and had an average age at death of 78.30 years. This underscores the role of the Mediterranean diet in reducing not only cardiovascular disease [30,31,32] but also the risk of cancer [2,3] and type 2 diabetes [33], the latter being a significant factor in cancer development due to insulin resistance and compensatory hyperinsulinemia [34].

### Strengths and Limitations

This study has several strengths, including its focus on a well-defined Italian Mediterranean population. We utilized standardized and validated questionnaires to gather comprehensive data on dietary, socioeconomic, and lifestyle factors, which helps minimize bias and confounding. The consistent dietary habits across age groups in this geographic region add robustness to our findings [35]. Additionally, the use of olives harvested before full ripening to produce high-quality extra virgin olive oil, rich in polyphenols, further strengthens this study’s validity. The long follow-up period of 17 years enhances our ability to identify significant long-term associations. However, this study has some limitations. The composition of the oil is not known. No official source reports its characteristics.

Despite the validation of the Food Frequency Questionnaire (FFQ) used, measurement errors are unavoidable. Dietary and laboratory data were collected only at baseline, meaning that changes in diet and biomarkers over time could influence the results. Nonetheless, evidence indicates that dietary patterns in adulthood generally remain stable, and the biomarkers tested showed minimal variation over time [35].

## 5. Conclusions

In this cohort of Italian Mediterranean adults, a higher consumption of polyphenol-rich EVOO was associated with a reduced risk of cancer and all-cause mortality, reflecting the benefits of a high-quality Mediterranean diet. The most pronounced benefits were observed with daily intakes of EVOO exceeding 50 g. The protective effect against cancer mortality can be attributed to its potential to reduce oxidative stress, inflammation, and induce beneficial epigenetic and metagenomic changes. Our findings underscore the importance of promoting daily EVOO consumption as a preventive dietary measure, particularly for individuals at increased risk of colorectal cancer. To further elucidate the role of EVOO and its impact on health, additional population-based studies and randomized clinical trials are needed to expand our understanding of this dietary intervention and its broader implications for cancer prevention and overall health.

## Figures and Tables

**Figure 1 cancers-16-03575-f001:**
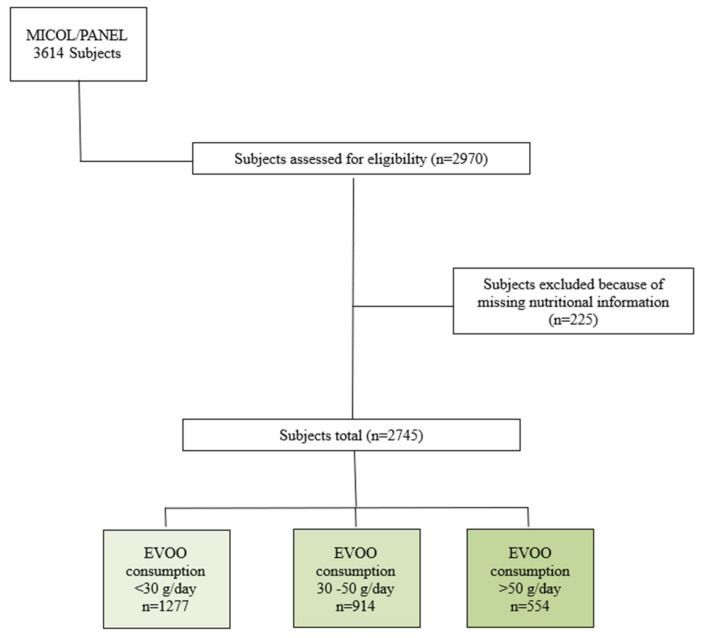
Flow chart.

**Figure 2 cancers-16-03575-f002:**
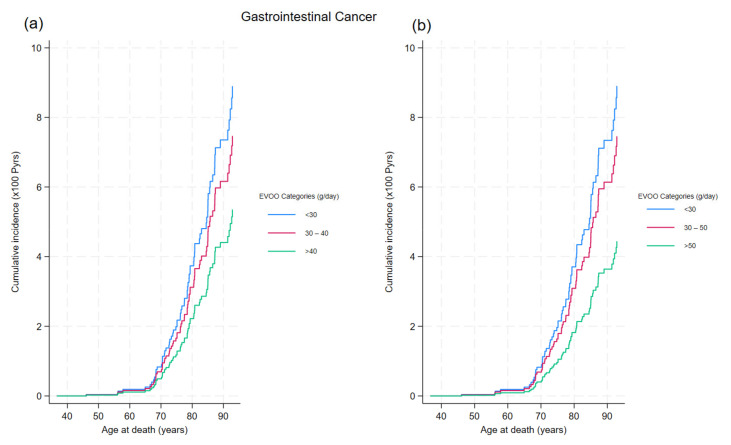
Cumulative incidence of gastrointestinal cancer mortality by EVOO consumption categories. (**a**) Cumulative incidence functions for gastrointestinal cancer mortality are plotted across three EVOO consumption categories: <30 g/day, 30–40 g/day, and >40 g/day. The graph illustrates the probability of death from gastrointestinal cancer over time, with the lines representing different EVOO consumption levels. (**b**) Cumulative incidence functions for gastrointestinal cancer mortality are shown for EVOO consumption categories: <30 g/day, 30–50 g/day, and >50 g/day. This plot highlights how varying levels of EVOO intake impact the probability of death from gastrointestinal cancer, with distinct lines for each consumption category.

**Table 1 cancers-16-03575-t001:** Participant characteristics by EVOO consumption categories in the MICOL/PANEL Studies (2005–2023).

	All Samples ^4^	EVOO Categories (g/Day) ^5^			*p*-Value ^6^
		<30	30–50	>50	
N (%) ^1^	2745 (100)	1277 (46.5)	914 (33.3)	554 (20.2)	
Sex ^1^					0.035
Female	1188 (43.3)	585 (49.2)	369 (31.1)	234 (19.7)	
Male	1557 (56.7)	692 (44.4)	545 (35.0)	320 (20.6)	
Age at enrollment (years) ^2^	54.73 (14.97)	49.24 (13.80)	57.16 (14.46)	63.37 (13.16)	<0.001
SBP (mmHg) ^2^	123.61 (19.81)	119.01 (18.74)	126.18 (20.04)	129.96 (19.34)	<0.001
DBP (mmHg) ^2^	74.80 (10.13)	74.15 (10.52)	75.29 (9.88)	75.47 (9.54)	0.007
Weight (kg) ^2^	75.15 (15.25)	74.58 (15.11)	75.35 (15.06)	76.13 (15.86)	0.12
BMI (kg/m^2^) ^2^	28.71 (5.17)	28.03 (4.91)	28.89 (5.04)	30.00 (5.69)	<0.001
Waist (cm) ^2^	92.99 (13.32)	90.93 (13.41)	93.61 (12.35)	96.74 (13.73)	<0.001
WHR ^2^	0.90 (0.08)	0.89 (0.08)	0.91 (0.08)	0.91 (0.07)	<0.001
Kcalories ^2^	2191.9 (701.1)	2085.75 (701.63)	2215.42 (685.30)	2397.19 (677.49)	0.20
Kcalories, no oil ^2^	1917.3 (699.2)	1938.85 (721.37)	1912.65 (689.39)	1875.61 (661.76)	0.60
TG (mmol/L) ^2^	128.99 (92.14)	127.53 (99.75)	131.46 (88.73)	128.30 (78.41)	0.40
TC (mmol/L) ^2^	199.50 (38.53)	199.46 (38.67)	200.59 (38.00)	197.77 (39.08)	0.18
HDL (mmol/L) ^2^	51.19 (13.97)	51.31 (14.55)	50.57 (13.11)	51.93 (13.97)	0.092
LDL (mmol/L) ^2^	122.54 (33.21)	122.92 (33.50)	123.64 (33.01)	119.86 (32.78)	0.038
Glucose (mmol/L) ^2^	109.16 (26.95)	107.79 (25.99)	110.03 (27.87)	110.89 (27.45)	0.90
ALT (U/L) ^2^	17.72 (15.36)	17.58 (13.42)	17.83 (14.96)	17.87 (19.64)	<0.001
AST (U/L) ^2^	12.91 (9.67)	12.24 (6.82)	13.13 (10.05)	14.10 (13.70)	<0.001
Age at death (years) ^3^	71.50 (58.89–82.10)	62.94 (55.99–75.27)	74.71 (63.26–82.94)	80.88 (71.93–86.32)	<0.001
Smoke ^1^					<0.001
Never/Former	2265 (82.5)	1012 (44.7)	765 (33.8)	488 (21.5)	
Current	480 (17.5)	265 (55.2)	149 (31.0)	66 (13.8)	
Observation time (years) ^3^	17.12 (17.03–18.28)	17.91 (17.12–18.18)	17.95 (16.97–18.44)	17.84 (12.47–18.33)	<0.001
Status ^1^					<0.001
Alive and/or censored	2052 (74.8)	1072 (52.2)	665 (32.4)	315 (15.4)	
Dead	693 (25.2)	205 (29.6)	249 (35.9)	239 (34.5)	
Causes of Death ^1^					<0.001
GC	87 (12.55)	31 (35.6)	35 (40.2)	21 (24.1)	
OCr	122 (17.60)	44 (36.1)	46 (37.7)	32 (26.2)	
DOC	484 (69.84)	130 (26.9)	168 (34.7)	186 (38.4)	
CCI ^3^	3.00 (1.00–5.00)	2.00 (1.00–4.00)	4.00 (2.00–5.00)	4.00 (3.00–6.00)	
rMED Score ^1^					<0.001
Low	631 (23.9)	395 (62.6)	167 (26.5)	69 (10.9)	
Medium	1.660 (63.0)	730 (44.0)	579 (34.9)	351 (21.1)	
High	345 (13.1)	118 (34.2)	127 (36.8)	100 (29.0)	
rMED, no Oil ^3^	7 (5–8)	7 (5–8)	7 (5–8)	7 (5–8)	0.089
Job ^1^					<0.001
Managers and Professionals	119 (4.3)	62 (52.1)	38 (31.9)	19 (16.0)	
Craft, Africultural, and Sales Workers	687 (25.0)	378 (55.0)	223 (32.5)	86 (12.5)	
Elementary Occupations	686 (25.0)	350 (51.0)	213 (31.0)	123 (17.9)	
Housewifes	312 (11.4)	148 (47.4)	102 (32.7)	62 (19.9)	
Pensioneers	899 (32.8)	313 (34.8)	328 (36.5)	258 (28.7)	
Jobless	42 (1.5)	26 (61.9)	10 (23.8)	6 (14.3)	
Education ^1^					<0.001
Primary School	921 (33.6)	374 (40.6)	316 (34.3)	231 (25.1)	
Secondary School	847 (30.9)	380 (44.9)	283 (33.4)	184 (21.7)	
High School	659 (24.0)	363 (55.1)	201 (30.5)	95 (14.4)	
Graduated	314 (11.5)	158 (50.3)	113 (36.0)	43 (13.7)	
Marital Status ^1^					<0.001
Single	299 (10.9)	135 (45.2)	108 (36.1)	56 (18.7)	
Married/Cohabiting	2123 (77.3)	1002 (47.2)	713 (33.6)	408 (19.2)	
Separated/Divorced	83 (3.0)	48 (57.8)	21 (25.3)	14 (16.9)	
Widower	240 (8.7)	92 (38.3)	72 (30.0)	76 (31.7)	
Diabetes ^1^					0.019
No	2472 (90.1)	1172 (47.4)	809 (32.7)	491 (19.9)	
Yes	273 (9.9)	105 (38.5)	105 (38.5)	63 (23,1)	
Dyslipidaemia ^1^					<0.001
No	2034 (74.1)	970 (53.3)	559 (30.7)	290 (15.9)	
Yes	711 (25.9)	307 (33.2)	355 (38.3)	264 (28,5)	
Hypertension ^1^					0.002
No	1819 (66.3)	985 (48.4)	646 (31.8)	403 (19.8)	
Yes	926 (33.7)	292 (41.1)	268 (37.7)	151 (21.2)	

DBP: Diastolic Blood Pressure; SBP: Systolic Blood Pressure; BMI: Body Mass Index; WHR: Waist-to-hip ratio; TG: Triglycerides; TC: Total Cholesterol; HDL: High-Density Lipoprotein Cholesterol; LDL: Low-Density Lipoprotein Cholesterol; ALT: Alanine Aminotransferase; AST: Aspartate Transaminase; GC: Gastrointestinal Cancer; OCr: Other Cancers; DOC: Other Causes of Death; CCI: Charlson Comorbidity Index; rMED: relative Mediterranean Scoring System. rMED no Oil: relative Mediterranean Diet Score without EVOO. Cells reporting subject characteristics contain ^1^ number (%), ^2^ mean ± (SD), ^3^ median (IQR), ^4^ percentages calculated per column, and ^5^ percentages calculated per row; ^6^ Kolmogorov test and Pearson’s chi-square tests were used to test differences between means and proportions, respectively.

**Table 2 cancers-16-03575-t002:** Distribution of food group consumption by EVOO intake categories (g/day).

	EVOO Categories (g/Day)			
	<30	30–50	>50	*p*-Value ^1^
N (%)	1277 (46.5)	914 (33.3)	554 (20.2)	
	Mean(SD)	Mean(SD)	Mean(SD)	
Cereals (g/day)	275.79 (145.49)	270.78 (137.20)	257.97 (129.91)	0.043
Vegetables (g/day)	241.14 (220.40)	248.36 (181.35)	266.24 (195.35)	0.052
Legumes (g/day)	33.69 (35.09)	31.92 (22.01)	34.60 (29.37)	0.21
Fish (g/day)	39.23 (34.15)	36.03 (27.37)	35.76 (29.37)	0.021
Total Meat (g/day)	87.09 (55.27)	74.24 (47.09)	71.56 (48.42)	<0.001
Dairy Products (g/day)	213.78 (151.72)	208.37 (148.75)	192.73 (148.86)	0.022
Fruits (g/day)	662.49 (565.79)	696.49 (550.22)	661.91 (537.32)	0.31
Wine (mL/day)	120.20 (181.89)	145.20 (186.72)	180.67 (228.98)	<0.001
Total Alcohol (mL/day)	160.83 (236.41)	179.61 (231.95)	206.52 (263.22)	<0.001

Cereals: Barley, Dry Pasta, Rice, Risottos, Pastina, Pizza, Focaccia, Bread, Biscuits, Pastries. Vegetables: Aubergines, Courgettes, Spinach, Cabbage, Cauliflower, Turnip Greens, Cucumbers, Watermelons, Green Beans, Fava Beans with Vegetables, Carrots, Artichokes, Peppers, Fennel, Celery, Swiss Chard, Chicory, Green Salad, Salad Tomatoes. Legumes: Broad Beans, Chickpeas, Beans, Lentils, Peas. Fish: Octopus, Cuttlefish, Squid, Shrimps, Anchovies, Sardines, Mussels, Mullet, Mackerel. Total Meat: Veal, Lamb, Horse, Rabbit, Chicken, Liver, Pork, Sausage, Mortadella, Cooked Ham, Raw Ham, Salami. Dairy Products: Whole Milk, Partially Skimmed Milk, Skimmed Milk, Yogurt, Grana, Parmesan, Caciottina, Stracchino, Fontina, Pecorino, Vacchino, Provolone, Caciocavallo, Formaggino, Belpaese, Gorgonzola, Scamorza, Swiss, Mozzarella, Ricotta. Fruit: Peaches, Watermelon, Apricots, Cherries, Grapes, White Melon, Apples, Pears, Citrus Fruits, Kiwis. rMED: relative Mediterranean Scoring System. ^1^ Kruskal–Wallis test).

**Table 3 cancers-16-03575-t003:** Hazard Ratios (HR) and Subdistribution Hazard Ratios (SHR) for Mortality by EVOO consumption categories (g/day).

	All Causes	GC	OCr	DOC
EVOO Categories (g/day)	HR (95% CI)	SHR (95% CI)	SHR (95% CI)	SHR (95% CI)
Model a				
<30	1.00	1.00	1.00	1.00
30–50	0.77 * (0.63; 0.94)	0.82 (0.50; 1.37)	0.81 (0.54; 1.24)	0.74 * (0.58; 0.95)
>50	0.80 * (0.65; 0.98)	0.39 * (0.21; 0.73)	0.50 * (0.31; 0.81)	1.02 (0.80; 1.29)
Model b	HR (95% CI)	SHR (95% CI)	SHR (95% CI)	SHR (95% CI)
<30	1.00	1.00	1.00	1.00
30–40	0.83 (0.67; 1.04)	0.74 (0.41; 1.35)	0.90 (0.56; 1.44)	0.76 * (0.58; 0.99)
>40	0.77 * (0.64; 0.93)	0.55 * (0.33; 0.93)	0.56 * (0.36; 0.85)	0.92 (0.73; 1.15)

* *p*-value < 0.05. Models a and b adjusted for sex (F vs. M), total calories without oil, Alanine Aminotransferase, waist circumference, Diastolic and Systolic Blood Pressure, relative Mediterranean Diet Score without EVOO, smoking, job, and Charlson Comorbidity Index. HR: Hazard Ratio; SHR: Subdistribution Hazard Ratio; GC: Gastrointestinal Cancer; OCr: Other Cancer Deaths; DOC: Deaths from Other Causes; EVOO.

## Data Availability

The data are available by contacting the corresponding author at the email address catia.bonfiglio@irccsdebellis.it.

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
