# Peer review of "Protective Effect of Extra Virgin Olive Oil on Cancers, Gastrointestinal Cancers, and All-Cause Mortality: A Competing Risk Analysis in a Southern Italian Cohort"

_cancers, 2024, doi:10.3390/cancers16213575_

Round 1

Reviewer 1 Report

Comments and Suggestions for Authors

This is a well-written manuscript and adds to the evidence about the effects of olive oil and the Mediterranean diet on health. 

Please report the people who volunteered for the research 'participants' rather than 'subjects'. The move towards doing research with participants rather than to them has grown and should be recognised. 

Some extra detail would be helpful:

Why did you choose EVOO? Olive oil per se has been shown to be protective.

How can you be sure this is what was consumed? How did you measure EVOO intake?

How were all the participants contacted? 

Was consent verbal or written?

How did you gain participation and consent of illiterate people?

How were participants interviewed, who interviewed, etc.

Given that those who consumed this highest amount of EVOO also consumed high levels of vegetables and legumes - also known to be protective against cancers, how can you be sure it was the EVOO? These are confounding factors and should be recognised as limitations.

Author Response

The attached answer

Reviewer 2 Report

Comments and Suggestions for Authors

The manuscript by Bofiglio et al. evaluated the association between Extra Virgin Olive Oil (EVOO) consumption and mortality risk in a cohort of Italian adults adhering to the Mediterranean diet. The authors observed that higher EVOO consumption was associated with significantly reducing cancer and all-cause mortality.

Comments

- How do the authors discriminate if the reduction of cancer and all-cause mortality is due to high EVOO consumption or Med Diet adherence?  

- Inclusion and exclusion criteria of enrolled patients should be better elucidated.

- The authors should evaluate the therapy of enrolled patients.

- What is the power of the study?

- Did the authors observed gender differences?

- Extra Virgin Olive Oil should be characterized. Did the enrolled patients consume EVOO cultivar with high content of polyphenols?   

- The authors should better explain the Food Frequency Questionnaire used. Have the authors evaluated the Mediterranean diet adherence with ad-hoc questionnaire?

Comments on the Quality of English Language

The manuscript should be edited to correct contextual and layout errors.

Author Response

The attached answer

Reviewer 3 Report

Comments and Suggestions for Authors

Thank you for the opportunity to review this paper

 This is a very  interesting review entitled:  Protective effect of extra virgin olive oil on cancers, gastrointestinal cancers, and All-cause Mortality: A Competing Risk Analysis in a Southern Italian Cohort

 Some topics in the article should be considered:

 Results

1.  Table 1 presents the anthropometric, demographic, hematological and biochemical characteristics of the participants descriptively in relation to the consumption of olive oil, but the effect of this on the markers presented is not presented. It would be interesting to make a statistical comparison taking into account the amount of olive oil consumption

2. The results should mention if there is any difference in the effect observed between men and women

3. It would be interesting to carry out a multi-factorial analysis to observe if risk factors for intestinal cancer such as the consumption of meat, dairy products or alcohol in combination with the consumption of olive oil have a less detrimental effect on this type of cancer

Limitations

In the limitations, it should be mentioned the fact that the quality of olive oil is not known with respect to its biocomponents; it is known from many studies based on evidence that the quantity and type of polyphenols contained can contribute specifically to the prevention of cancer. These factors depend on the region in which the olives are harvested and the type of oil extraction.

Table S2 please add the explanation for abbreviation GC, OCr and DOC

Author Response

The attached answer

Reviewer 4 Report

Comments and Suggestions for Authors

Dear Authors,

Thank you for your work over time and on a substantial amount of data.

In the last decades, In Italy, as well as in all the countries of the Mediterranean basin, we are witnessing in a progressive shift away from the Mediterranean model, which is increasingly being replaced by the consumption of ultra-processed foods. Emphasizing how EVOO, one of the pillars of the Mediterranean diet, exerts such an important role in reducing the risk of cancer and all-cause mortality is a key issue.

In studies focusing on dietary habits, however, it is not easy to isolate the effect of a single component, as EVOO, and to exclude the synergistic action of a multitude of phytocompounds, which are particularly abundant in vegetables and legumes. In analysing the data, did you try to subdivide the population according to MD adherence and, within the same category (low, moderate and high), to test the risk of cancer/mortality in the population consuming >50g/day vs 30g/day? This further investigation would perhaps allow even greater emphasis on the positive effect of EVOO.

Minor:

How was the analysis of the responses provided by the FFQ conducted? Did you use bromatological software?

With which instrument was evaluated the adherence to MD? In table 2 the description of rMED as median is not very suggestive, could you use another value, e.g. “mode”?

The categories presented in Table 1 (N, Sex, Smoke, Status, and so on...) are not well highlighted. To make the table easier to read, I suggest using boldface.

Thank you.

Author Response

The attached answer
